# Soft Actor-Critic:
## Off-Policy Maximum Entropy Deep Reinforcement Learning with a Stochastic Actor

## Abstract

Model-free deep reinforcement learning (RL) algorithms have been demonstrated on a range of challenging decision making and control tasks. However, these methods typically suffer from two major challenges: very high sample complexity and brittle convergence properties, which necessitate meticulous hyperparameter tuning. Both of these challenges severely limit the applicability of such methods to complex, real-world domains. In this paper, we propose soft actor-critic, an off-policy actor-critic deep RL algorithm based on the maximum entropy reinforcement learning framework. In this framework, the actor aims to maximize expected reward while also maximizing entropy—that is, succeed at the task while acting as randomly as possible. Prior deep RL methods based on this framework have been formulated as either off-policy Q-learning, or on-policy policy gradient methods. By combining off-policy updates with a stable stochastic actor-critic formulation, our method achieves state-of-the-art performance on a range of continuous control benchmark tasks, outperforming prior on-policy and off-policy methods. Furthermore, we demonstrate that, in contrast to other off-policy algorithms, our approach is very stable, achieving very similar performance across different random seeds.

## 1 Introduction

Model-free deep reinforcement learning (RL) algorithms have been applied in a range of challenging domains, from games (Mnih et al., 2013; Silver et al., 2016) to robotic control (Schulman et al., 2015). The combination of RL and high-capacity function approximators such as neural networks holds the promise of automating a wide range of decision making and control tasks, but widespread adoption of these methods in real-world domains has been hampered by two major challenges. First, model-free deep RL methods are notoriously expensive in terms of their sample complexity. Even relatively simple tasks can require millions of steps of data collection, and complex behaviors with high-dimensional observations might need substantially more. Second, these methods are often brittle with respect to their hyperparameters: learning rates, exploration constants, and other settings must be set carefully for different problem settings to achieve good results. Both of these challenges severely limit the applicability of model-free deep RL to real-world tasks.

One cause for the poor sample efficiency of deep RL methods is on-policy learning: some of the most commonly used deep RL algorithms, such as TRPO (Schulman et al., 2015) or A3C (Mnih et al., 2016), require new samples to be collected for each gradient step on the policy. This quickly becomes extravagantly expensive, as the number of gradient steps to learn an effective policy increases with task complexity. Off-policy algorithms instead aim to reuse past experience. This is not directly feasible with conventional policy gradient formulations (Schulman et al., 2015; Mnih et al., 2016), but is relatively straightforward for Q-learning based methods (Mnih et al., 2015). Unfortunately, the combination of off-policy learning and high-dimensional, nonlinear function approximation with neural networks presents a major challenge for stability and convergence (Bhatnagar et al., 2009). This challenge is further exacerbated in continuous state and action spaces, where a separate actor network is typically required to perform the maximization in Q-learning. A commonly used algorithm in such settings, deep deterministic policy gradient (DDPG) (Lillicrap et al., 2015), provides for sample-efficient learning, but is notoriously challenging to use due to its extreme brittleness and hyperparameter sensitivity (Duan et al., 2016; Henderson et al., 2017).

We explore how to design an efficient and stable model-free deep RL algorithm for continuous state and action spaces. To that end, we draw on the maximum entropy framework, which augments the standard maximum reward reinforcement learning objective with an entropy maximization term (Ziebart et al., 2008; Toussaint, 2009; Rawlik et al., 2012; Fox et al., 2016; Haarnoja et al., 2017). Maximum entropy reinforcement learning alters the RL objective, though the original objective can be recovered by using a temperature parameter (Haarnoja et al., 2017). More importantly, the maximum entropy formulation provides a substantial improvement in exploration and robustness: as discussed by Ziebart (2010), maximum entropy policies are robust in the face of modeling and estimation errors, and as demonstrated by Haarnoja et al. (2017), they improve exploration by acquiring diverse behaviors. Prior work has proposed model-free deep RL algorithms that perform on-policy learning with entropy maximization (O'Donoghue et al., 2016), as well as off-policy methods based on soft Q-learning and its variants (Schulman et al., 2017; Nachum et al., 2017a; Haarnoja et al., 2017). However, the on-policy variants suffer from poor sample complexity for the reasons discussed above, while the off-policy variants require complex approximate inference procedures in continuous action spaces.

In this paper, we demonstrate that we can devise an off-policy maximum entropy actor-critic algorithm, which we call soft actor-critic, which provides for both sample-efficient learning and stability. This algorithm extends readily to very complex, high-dimensional tasks, such as the 21-DoF Humanoid benchmark (Duan et al., 2016), where off-policy methods such as DDPG typically struggle to obtain good results (Gu et al., 2016), while avoiding the complexity and potential instability associated with approximate inference in prior off-policy maximum entropy algorithms based on soft Q-learning (Haarnoja et al., 2017). In particular, we present a novel convergence proof for policy iteration in the maximum entropy framework. We then introduce a new algorithm based on an approximation to this procedure that can be practically implemented with deep neural networks, which we call soft actor-critic. We present empirical results that show that soft actor-critic attains a substantial improvement in both performance and sample efficiency over both off-policy and on-policy prior methods.

## 2 RELATED WORK

Our soft actor-critic algorithm incorporates three key ingredients: an actor-critic architecture with separate policy and value function networks, an off-policy formulation that enables reuse of previously collected data for efficiency, and entropy maximization to enable stability and exploration. We review prior works that draw on some of these ideas in this section. Actor-critic algorithms are typically derived starting from policy iteration, which alternates between *policy evaluation*—computing the value function for a policy—and *policy improvement*—using the value function to obtain a better policy (Barto et al., 1983; Sutton & Barto, 1998). In large-scale reinforcement learning problems, it is typically impractical to run either of these steps to convergence, and instead the value function and policy are optimized jointly. In this case, the policy is referred to as the actor, and the value function as the critic. Many actor-critic algorithms build on the standard, on-policy policy gradient formulation to update the actor (Peters & Schaal, 2008), and many of them also consider the entropy of the policy, but instead of maximizing the entropy, they use it as an regularizer (Schulman et al., 2015; Mnih et al., 2016; Gruslys et al., 2017). This tends to improve stability, but results in very poor sample complexity.

There have been efforts to increase the sample efficiency while retaining the robustness properties by incorporating off-policy samples and by using higher order variance reduction techniques (O'Donoghue et al., 2016; Gu et al., 2016). However, fully off-policy algorithms still attain better efficiency. A particularly popular off-policy actor-critic method, DDPG Lillicrap et al. (2015), which is a deep variant of the deterministic policy gradient (Silver et al., 2014) algorithm, uses a Q-function estimator to enable off-policy learning, and a deterministic actor that maximizes this Q-function. As such, this method can be viewed both as a deterministic actor-critic algorithm and an approximate Q-learning algorithm. Unfortunately, the interplay between the deterministic actor network and the Q-function typically makes DDPG extremely difficult to stabilize and brittle to hyperparameter settings (Duan et al., 2016; Henderson et al., 2017). As a consequence, it is difficult to extend DDPG to very complex, high-dimensional tasks, and on-policy policy gradient methods still tend to produce the best results in such settings (Gu et al., 2016). Our method instead combines off-policy actor-critic training with a *stochastic* actor, and further aims to maximize the entropy of this actor with an entropy maximization objective. We find that this actually results in

a substantially more stable and scalable algorithm that, in practice, exceeds both the efficiency and final performance of DDPG.

Maximum entropy reinforcement learning optimizes policies to maximize both the expected return and the expected entropy of the policy. This framework has been used in many contexts, from inverse reinforcement learning (Ziebart et al., 2008) to optimal control (Todorov, 2008; Toussaint, 2009; Rawlik et al., 2012). In guided policy search (Levine & Koltun, 2013), maximum entropy distribution is used to guide policy learning towards high-reward regions. More recently, several papers have noted the connection between Q-learning and policy gradient methods in the framework of maximum entropy learning (O'Donoghue et al., 2016; Haarnoja et al., 2017; Nachum et al., 2017a; Schulman et al., 2017). While most of the prior works assume a discrete action space, Nachum et al. (2017b) approximate the maximum entropy distribution with a Gaussian and Haarnoja et al. (2017) with a sampling network trained to draw samples from the optimal policy. Although the soft Q-learning algorithm proposed by Haarnoja et al. (2017) has a value function and actor network, it is not a true actor-critic algorithm: the Q-function is estimating the optimal Q-function, and the actor does not directly affect the Q-function except through the data distribution. Hence, Haarnoja et al. (2017) motivates the actor network as an approximate sampler, rather than the actor in an actor-critic algorithm. Crucially, the convergence of this method hinges on how well this sampler approximates the true posterior. In contrast, we prove that our method converges to the optimal policy from a given policy class, regardless of the policy parameterization. Furthermore, these previously proposed maximum entropy methods generally do not exceed the performance of state-of-the-art off-policy algorithms, such as DDPG, when learning from scratch, though they may have other benefits, such as improved exploration and ease of finetuning. In our experiments, we demonstrate that our soft actor-critic algorithm does in fact exceed the performance of state-of-the-art off-policy deep RL methods by a wide margin.

## 3 PRELIMINARIES

In this section, we introduce notation and summarize the standard and maximum entropy reinforcement learning frameworks.

### 3.1 NOTATION

We address policy learning in continuous action spaces. To that end, we consider infinite-horizon Markov decision processes (MDP), defined by the tuple $(\mathcal{S}, \mathcal{A}, p_{\mathbf{s}}, r)$, where the state space $\mathcal{S}$ and the action space $\mathcal{A}$ are assumed to be continuous, and the unknown state transition probability $p_{\mathbf{s}} : \mathcal{S} \times \mathcal{S} \times \mathcal{A} \to [0, \infty)$ represents the probability density of the next state $\mathbf{s}_{t+1} \in \mathcal{S}$ given the current state $\mathbf{s}_t \in \mathcal{S}$ and action $\mathbf{a}_t \in \mathcal{A}$. The environment emits a bounded reward $r : \mathcal{S} \times \mathcal{A} \to [r_{\min}, r_{\max}]$ on each transition, which we will abbreviate as $r_t \triangleq r(\mathbf{s}_t, \mathbf{a}_t)$ to simplify notation. We will also use $\rho_\pi(\mathbf{s}_t)$ and $\rho_\pi(\mathbf{s}_t, \mathbf{a}_t)$ to denote the state and state-action marginals of the trajectory distribution induced by a policy $\pi(\mathbf{a}_t|\mathbf{s}_t)$.

### 3.2 MAXIMUM ENTROPY REINFORCEMENT LEARNING

The standard objective used in reinforcement learning is to maximize the expected sum of rewards $\sum_t \mathbb{E}_{(\mathbf{s}_t, \mathbf{a}_t) \sim \rho_\pi} [r_t]$. We will consider a more general maximum entropy objective (see e.g. (Ziebart, 2010)), which favors stochastic policies by augmenting the objective with the expected entropy of the policy over $\rho_\pi(\mathbf{s}_t)$:

$$J(\pi) = \sum_{t=0}^{T-1} \mathbb{E}_{(\mathbf{s}_t, \mathbf{a}_t) \sim \rho_\pi} \left[ r(\mathbf{s}_t, \mathbf{a}_t) + \alpha \mathcal{H}(\pi(\,\cdot\,|\mathbf{s}_t)) \right]. \tag{1}$$

The temperature parameter $\alpha$ determines the relative importance of the entropy term against the reward, and thus controls the stochasticity of the optimal policy. The maximum entropy objective differs from the standard maximum expected reward objective used in conventional reinforcement learning, though the conventional objective can be recovered in the limit as $\alpha \to 0$. For the rest of this paper, we will omit writing the temperature explicitly, as it can always be subsumed into the reward by scaling it by $\alpha^{-1}$. The maximum entropy objective has a number of conceptual and practical advantages. First, the policy is incentivized to explore more widely, while giving up on clearly unpromising avenues. Second, the policy can capture multiple modes of near-optimal

behavior. In particular, in problem settings where multiple actions seem equally attractive, the policy will commit equal probability mass to those actions. Lastly, prior work has observed substantially improved exploration from this objective (Haarnoja et al., 2017; Schulman et al., 2017), and in our experiments, we observe that it considerably improves learning speed over state-of-art methods that optimize the conventional objective function. We can extend the objective to infinite horizon problems by introducing a discount factor $\gamma$ to ensure that the sum of expected rewards and entropies is finite. Writing down the precise maximum entropy objective for the infinite horizon discounted case is more involved (Thomas, 2014) and is deferred to Appendix A.

Prior methods have proposed directly solving for the optimal Q-function, from which the optimal policy can be recovered (Ziebart et al., 2008; Fox et al., 2016; Haarnoja et al., 2017). In the next section, we will discuss how we can devise a soft actor-critic algorithm through a policy iteration formulation, where we instead evaluate the Q-function of the current policy and update the policy through an *off-policy* gradient update. Though such algorithms have previously been proposed for conventional reinforcement learning, our method is, to our knowledge, the first off-policy actor-critic method in the maximum entropy reinforcement learning framework.

## 4    FROM SOFT POLICY ITERATION TO SOFT ACTOR-CRITIC

Our off-policy soft actor-critic algorithm can be derived starting from a novel maximum entropy variant of the policy iteration method. In this section, we will first present this derivation, verify that the corresponding algorithm converges to the optimal policy from its density class, and then present a practical deep reinforcement learning algorithm based on this theory.

### 4.1    DERIVATION OF SOFT POLICY ITERATION

We will begin by deriving soft policy iteration, a general algorithm for learning optimal maximum entropy policies that alternates between policy evaluation and policy improvement in the maximum entropy framework. Our derivation is based on a tabular setting, to enable theoretical analysis and convergence guarantees, and we extend this method into the general continuous setting in the next section. We will show that soft policy iteration converges to the optimal policy within a set of policies which might correspond, for instance, to a set of parameterized densities.

In the policy evaluation step of soft policy iteration, we wish to compute the value of a policy $\pi$ according to the maximum entropy objective in Equation 1. For a fixed policy, the soft Q-value can be computed iteratively, by starting from any function $Q : \mathcal{S} \times \mathcal{A} \to \mathbb{R}$ and iteratively applying a modified version of the Bellman backup operator $\mathcal{T}^\pi$ defined by

$$\mathcal{T}^\pi Q \triangleq r(\mathbf{s}_t, \mathbf{a}_t) + \gamma \mathbb{E}_{\mathbf{s}_{t+1} \sim p_\mathbf{s}} \left[ V(\mathbf{s}_{t+1}) \right], \tag{2}$$

where

$$V(\mathbf{s}_t) = \mathbb{E}_{\mathbf{a}_t \sim \pi} \left[ Q(\mathbf{s}_t, \mathbf{a}_t) - \log \pi(\mathbf{a}_t | \mathbf{s}_t) \right] \tag{3}$$

is the soft state value function. We can find the soft value of an arbitrary policy $\pi$ by repeatedly applying $\mathcal{T}^\pi$ as formalized in Lemma 1.

**Lemma 1** (Soft Policy Evaluation). *Consider the soft Bellman backup operator $\mathcal{T}^\pi$ in Equation 2 and a mapping $Q^0 : \mathcal{S} \times \mathcal{A} \to \mathbb{R}$, and define $Q^{k+1} = \mathcal{T}^\pi Q^k$. Then the sequence $Q^k$ will converge to the soft Q-value of $\pi$ as $k \to \infty$.*

*Proof.* See Appendix B.1.                                                                                                                □

In the policy improvement step, we update the policy towards the exponential of the new Q-function. This particular choice of update can be guaranteed to result into an improved policy in terms of its soft value, as we show in this section. Since in practice we prefer policies that are tractable, we will additionally restrict the policy to some set of policies $\Pi$, which can correspond, for example, to a parameterized family of distributions such as Gaussians. To account for the constraint that $\pi \in \Pi$, we project the improved policy into the desired set of policies. While in principle we could choose any projection, it will turn out to be convenient to use the information projection defined in terms of the Kullback-Leibler divergence. In the other words, in the policy improvement step, for each state, we update the policy according to

$$\pi_{\text{new}} = \arg \min_{\pi' \in \Pi} D_{\text{KL}} \left( \pi'(\,\cdot\, | \mathbf{s}_t) \, \| \, \exp \left( Q^{\pi_{\text{old}}}(\mathbf{s}_t, \,\cdot\,) - \log Z^{\pi_{\text{old}}}(\mathbf{s}_t) \right) \right). \tag{4}$$

The partition function $Z^{\pi_{\text{old}}}(\mathbf{s}_t)$ normalizes the second KL argument, and while it is intractable in general, it does not contribute to the gradient with respect to the new policy and can thus be ignored as noted in the next section. For this choice of projection, we can show that the new, projected policy has a higher value than the old policy with respect to the objective in Equation 1. We formalize this result in Lemma 2.

**Lemma 2** (Soft Policy Improvement). *Let $\pi_{\text{old}} \in \Pi$ and let $\pi_{\text{new}}$ be the optimizer of the minimization problem defined in Equation 4. Then $Q^{\pi_{\text{new}}}(\mathbf{s}_t, \mathbf{a}_t) \geq Q^{\pi_{\text{old}}}(\mathbf{s}_t, \mathbf{a}_t)$ for all $(\mathbf{s}_t, \mathbf{a}_t) \in \mathcal{S} \times \mathcal{A}$.*

*Proof.* See Appendix B.2. □

The full soft policy iteration algorithm alternates between the soft policy evaluation and the soft policy improvement steps, and it will provably converge to the optimal maximum entropy policy among the policies in $\Pi$ (Theorem 1). Although this algorithm will provably find the optimal solution, we can perform it in its exact form only in the tabular case. Therefore, we will next approximate the algorithm for continuous domains, where we need to rely on a function approximator to represent the Q-values, and running the two steps until convergence would be computationally too expensive. The approximation gives rise to a new practical algorithm, called soft actor-critic.

**Theorem 1** (Soft Policy Iteration). *Repeated application of soft policy evaluation and soft policy improvement to any $\pi \in \Pi$ converges to a policy $\pi^*(\cdot|\mathbf{s}_t)$ such that $Q^{\pi^*}(\mathbf{s}_t, \mathbf{a}_t) > Q^{\pi}(\mathbf{s}_t, \mathbf{a}_t)$ for all $\pi \in \Pi$, $\pi \neq \pi^*$ and for all $(\mathbf{s}_t, \mathbf{a}_t) \in \mathcal{S} \times \mathcal{A}$.*

*Proof.* See Appendix B.3. □

### 4.2 SOFT ACTOR-CRITIC

As discussed above, large, continuous domains require us to derive a practical approximation to soft policy iteration. To that end, we will use function approximators for both the Q-function and policy, and instead of running evaluation and improvement to convergence, alternate between optimizing both networks with stochastic gradient descent. For the remainder of this paper, we will consider a parameterized state value function $V_\psi(\mathbf{s}_t)$, soft Q-function $Q_\theta(\mathbf{s}_t, \mathbf{a}_t)$, and a tractable policy $\pi_\phi(\mathbf{a}_t|\mathbf{s}_t)$. The parameters of these networks are $\psi$, $\theta$, and $\phi$. In the following, we will derive update rules for these parameter vectors.

The state value function approximates the soft value. There is no need in principle to include a separate function approximator for the state value, since it is related to the Q-function and policy according to Equation 3. This quantity can be estimated from a single action sample from the current policy without introducing a bias, but in practice, including a separate function approximator for the soft value can stabilize training—and as discussed later, can be used as a state-dependent baseline for learning the policy—and is convenient to train simultaneously with the other networks. The soft value function is trained to minimize the squared residual error

$$J_V(\psi) = \mathbb{E}_{\mathbf{s}_t \sim \mathcal{D}} \left[ \frac{1}{2} \left( V_\psi(\mathbf{s}_t) - \mathbb{E}_{\mathbf{a}_t \sim \pi_\phi} \left[ Q_\theta(\mathbf{s}_t, \mathbf{a}_t) - \log \pi_\phi(\mathbf{a}_t|\mathbf{s}_t) \right] \right)^2 \right], \tag{5}$$

where $\mathcal{D}$ is the distribution of previously sampled states and actions, or a replay buffer. The gradient of Equation 5 can be estimated with an unbiased estimator

$$\hat{\nabla}_\psi J_V(\psi) = \nabla_\psi V_\psi(\mathbf{s}_t) \left( V_\psi(\mathbf{s}_t) - Q_\theta(\mathbf{s}_t, \mathbf{a}_t) + \log \pi_\phi(\mathbf{a}_t|\mathbf{s}_t) \right), \tag{6}$$

where the actions are sampled according to the current policy, instead of the replay buffer. The soft Q-function parameters can be trained to minimize the soft Bellman residual

$$J_Q(\theta) = \mathbb{E}_{(\mathbf{s}_t, \mathbf{a}_t) \sim \mathcal{D}} \left[ \frac{1}{2} \left( Q_\theta(\mathbf{s}_t, \mathbf{a}_t) - \left( r(\mathbf{s}_t, \mathbf{a}_t) + \gamma \, \mathbb{E}_{\mathbf{s}_{t+1} \sim p_{\mathbf{s}}} \left[ V_{\bar{\psi}}(\mathbf{s}_{t+1}) \right] \right) \right)^2 \right], \tag{7}$$

which again can be optimized with stochastic unbiased gradients

$$\hat{\nabla}_\theta J_Q(\theta) = \nabla_\theta Q_\theta(\mathbf{a}_t, \mathbf{s}_t) \left( Q_\theta(\mathbf{s}_t, \mathbf{a}_t) - r(\mathbf{s}_t, \mathbf{a}_t) - \gamma V_{\bar{\psi}}(\mathbf{s}_{t+1}) \right). \tag{8}$$

The update makes use of a target value network $V_{\bar{\psi}}$ where $\bar{\psi}$ is an exponentially moving average of the value network weights, which has been shown to stabilize training (Mnih et al., 2015), although

we found soft actor-critic to be able to learn robustly also in the absense of a target network. Finally, the policy parameters can be learned by directly minimizing the KL-divergence in Equation 4, which we reproduce here in parametrized form for completeness

$$J_\pi(\phi) = \mathrm{D_{KL}}\left(\pi_\phi(\,\cdot\,|\mathbf{s}_t) \,\|\, \exp\left(Q_\theta(\mathbf{s}_t, \cdot) - \log Z_\theta(\mathbf{s}_t)\right)\right). \tag{9}$$

There are several options for minimizing $J_\pi$, depending on the choice of the policy class. For simple distributions, such as Gaussians, we can use the reparametrization trick, which allows us to back-propagate the gradient from the critic network and leads to a DDPG-style estimator. However, if the policy depends on discrete latent variables, such as is the case for mixture models, the reparametrization trick cannot be used. We therefore propose to use a likelihood ratio gradient estimator:

$$\nabla_\phi J_\pi(\phi) = \mathbb{E}_{\mathbf{a}_t \sim \pi_\phi}\left[\nabla_\phi \log \pi_\phi(\mathbf{a}_t|\mathbf{s}_t)\left(\log \pi_\phi(\mathbf{a}_t|\mathbf{s}_t) + 1 - Q_\theta(\mathbf{s}_t, \mathbf{a}_t) + \log Z_\theta(\mathbf{s}_t) + b(\mathbf{s}_t)\right)\right], \tag{10}$$

where $b(\mathbf{s}_t)$ is a state-dependent baseline (Peters & Schaal, 2008). We can approximately center the learning signal and eliminate the intractable log-partition function by choosing $b(\mathbf{s}_t) = V_\psi(\mathbf{s}_t) - \log Z_\theta(\mathbf{s}_t) - 1$, which yields the final gradient estimator

$$\hat{\nabla}_\phi J_\pi(\phi) = \nabla_\phi \log \pi_\phi(\mathbf{a}_t|\mathbf{s}_t)\left(\log \pi_\phi(\mathbf{a}_t|\mathbf{s}_t) - Q_\theta(\mathbf{s}_t, \mathbf{a}_t) + V_\psi(\mathbf{s}_t)\right). \tag{11}$$

The complete algorithm is described in Algorithm 1. The method alternates between collecting experience from the environment with the current policy and updating the function approximators using the stochastic gradients on batches sampled from a replay buffer. In practice we found a combination of a single environment step and multiple gradient steps to work best (see Appendix E). Using off-policy data from a replay buffer is feasible because both value estimators and the policy can be trained entirely on off-policy data. The algorithm is agnostic to the parameterization of the policy, as long as it can be evaluated for any arbitrary state-action tuple. We will next suggest a practical parameterization for the policy, based on Gaussian mixtures.

---

**Algorithm 1:** Soft Actor-Critic

1  Initialize parameter vectors $\psi, \bar{\psi}, \theta, \phi$.
2  **for** *each iteration* **do**
3      **for** *each environment step* **do**
4          $\mathbf{a}_t \sim \pi_\phi(\mathbf{a}_t|\mathbf{s}_t)$
5          $\mathbf{s}_{t+1} \sim p_\mathbf{s}(\mathbf{s}_{t+1}|\mathbf{s}_t, \mathbf{a}_t)$
6          $\mathcal{D} \leftarrow \mathcal{D} \cup \{(\mathbf{s}_t, \mathbf{a}_t, r(\mathbf{s}_t, \mathbf{a}_t), \mathbf{s}_{t+1})\}$.
7      **end**
8      **for** *each gradient step* **do**
9          $\psi \leftarrow \psi - \lambda_V \hat{\nabla}_\psi J_V(\psi)$
10         $\theta \leftarrow \theta - \lambda_Q \hat{\nabla}_\theta J_Q(\theta)$
11         $\phi \leftarrow \phi - \lambda_\pi \hat{\nabla}_\phi J_\pi(\phi)$
12         $\bar{\psi} \leftarrow \tau\psi + (1-\tau)\bar{\psi}$
13     **end**
14 **end**

---

### 4.3 Soft Actor-Critic with Gaussian Mixtures

Although we could use a simple policy represented by a Gaussian, as is common in prior work, the maximum entropy framework aims to maximize the randomness of the learned policy. Therefore, a more expressive but still tractable distribution can endow our method with more effective exploration and robustness, which are the typically cited benefits of entropy maximization (Ziebart, 2010). To that end, we propose a practical multimodal representation based on a mixture of $K$ Gaussians. This can approximate any distribution to arbitrary precision as $K \to \infty$, but even for practical numbers of mixture elements, it can provide a very expressive distribution in medium-dimensional action spaces. Although the complexity of evaluating or sampling from the resulting distribution scales linearly in $K$, our experiments indicates that a relatively small number of mixture components, typically just two or four, is sufficient to achieve high performance, thus making the algorithm scalable to complex domains with high-dimensional action spaces.

We define the policy as

$$\pi_\phi(\mathbf{a}_t|\mathbf{s}_t) = \frac{1}{\sum_i w_i^\phi} \sum_{i=1}^{K} w_i^\phi(\mathbf{s}_t)\mathcal{N}\left(\mathbf{a}_t; \mu_i^\phi(\mathbf{s}_t), \Sigma_i^\phi(\mathbf{s}_t)\right), \tag{12}$$

where $w_i^\phi$, $\mu_i^\phi$, $\Sigma_i^\phi$ are the unnormalized mixture weights, means, and covariances, respectively, which all can depend on $\mathbf{s}_t$ in complex ways if expressed as neural networks. We also apply a squashing function to limit the actions to a bounded interval as explained in Appendix C. Note that, in contrast to soft Q-learning (Haarnoja et al., 2017), our algorithm does *not* assume that the policy can accurately approximate the optimal exponentiated Q-function distribution. The convergence result for soft policy iteration holds even for policies that are restricted to a policy class, in contrast to prior methods of this type.

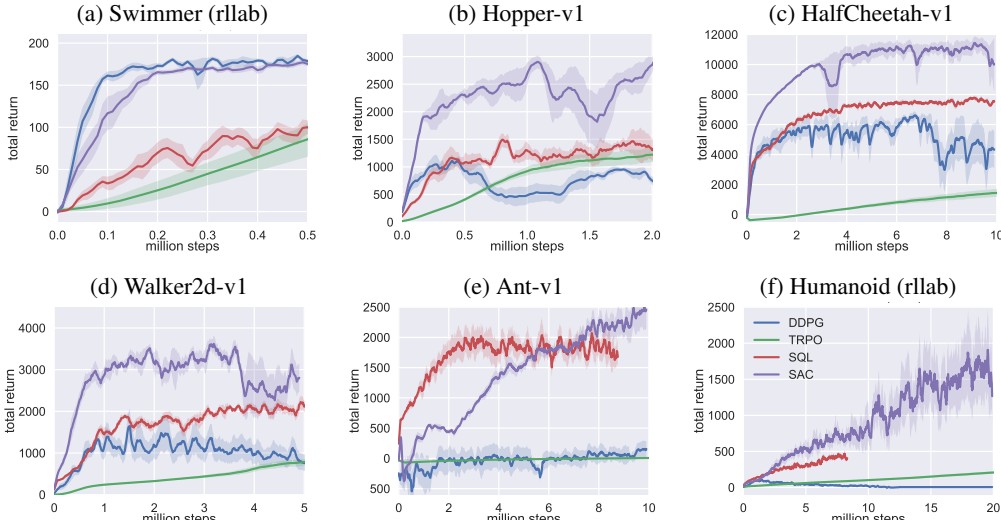

Figure 1: Training curves on continuous control benchmarks. Note that SAC performs consistently across all tasks attaining the highest score compared to both on-policy and off-policy methods in all benchmark tasks.

## 5 EXPERIMENTS

The goal of our experimental evaluation is to understand how the sample complexity and stability of our method compares with prior off-policy and on-policy deep reinforcement learning algorithms. To that end, we evaluate on a range of challenging continuous control tasks from the OpenAI gym benchmark suite (Brockman et al., 2016). For the Swimmer and Humanoid environments, we use the rllab implementations (Duan et al., 2016), which have more well-behaved observation spaces. Although the easier tasks in this benchmark suite can be solved by a wide range of different algorithms, the more complex benchmarks, such as the 21-dimensional Humanoid, are exceptionally difficult to solve with off-policy algorithms (Duan et al., 2016). The stability of the algorithm also plays a large role in performance: easier tasks make it more practical to tune hyperparameters to achieve good results, while the already narrow basins of effective hyperparameters become prohibitively small for the more sensitive algorithms on the most high-dimensional benchmarks, leading to poor performance (Gu et al., 2016).

In our comparisons, we compare to trust region policy optimization (TRPO) (Schulman et al., 2015), a stable and effective on-policy policy gradient algorithm; deep deterministic policy gradient (DDGP) (Lillicrap et al., 2015), an algorithm that is regarded as one of the more efficient off-policy deep RL methods (Duan et al., 2016); as well as soft Q-learning (SQL) (Haarnoja et al., 2017), an off-policy algorithm for learning maximum entropy policies. Although DDPG is very efficient, it is also known to be more sensitive to hyperparameter settings, which limits its effectiveness on complex tasks (Gu et al., 2016; Henderson et al., 2017). Figure 1 shows the total average reward of evaluation rollouts during training for the various methods. Exploration noise was turned off for evaluation for DDPG and TRPO. For maximum entropy algorithms, which do not explicitly inject exploration noise, we either evaluated with the exploration noise (SQL) or approximate the maximum a posteriori action by choosing the mean of the mixture component that achieves the highest Q-value (SAC).

### 5.1 COMPARATIVE EVALUATION

**Performance across tasks.** The results show that, overall, SAC substantially outperforms DDPG on all of the benchmark tasks in terms of final performance, and learns faster than any of the baselines in most environments. The only exceptions are Swimmer, where DDPG is slightly faster, and Ant-v1, where SQL is initially faster, but plateaus at a lower final performance. On the hardest tasks, Ant-v1 and Humanoid (rllab), DDPG is unable to make any progress, a result that is corroborated by prior work (Gu et al., 2016; Duan et al., 2016). SAC still learns substantially faster than TRPO on these tasks, likely as a consequence of improved stability. Poor stability is exacerbated by high-dimensional and complex tasks. Even though TRPO does not make perceivable progress for some tasks within the range depicted in the figures, it will eventually solve all of them after substantially more iterations. The quantitative results attained by SAC in our experiments also compare very favorably to results reported by other methods in prior work (Duan et al., 2016; Gu et al., 2016;

Henderson et al., 2017), indicating that both the sample efficiency and final performance of SAC on these benchmark tasks exceeds the state of the art. All hyperparameters used in this experiment for SAC are listed in Appendix E.

**Sensitivity to random seeds.** In Figure 2, we show the performance for multiple individual seeds on the HalfCheetah-v1 benchmark of DDPG, TRPO, and SAC (Figure 1 shows the average over seeds). The individual seeds attain much more consistent performance with SAC, while DDPG exhibits very high variability across seeds, indicating substantially worse stability. TRPO is unable to make progress within the first 1000 episodes. For SAC, we performed 16 gradient steps between each environment step, which becomes prohibitively slow in terms of wall clock time for longer experiments (in Figure 1 we used we used 4 gradient steps).

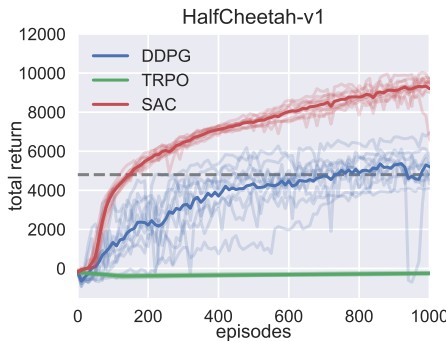

Figure 2: Performance of individual random seeds on the HalfCheetah-v1 benchmark.

## 5.2 ABLATION STUDY

The results in the previous section suggest that algorithms based on the maximum entropy principle can outperform conventional RL methods on challenging tasks such as the Ant and Humanoid. In this section, we further examine which particular components of SAC are important for good performance. To that end, we consider several ablations to pinpoint the most essential differences between our method and standard RL methods that do not maximize entropy. We use DDPG as the representative non-entropy-maximizing algorithm, since it is the most structurally similar to SAC. This section examines specific algorithm components, while a study of the sensitivity to hyperparameter settings—most importantly the reward scale and the number of gradient steps per environment step—is deferred to Appendix D.

**Importance of entropy maximization.** The main difference of SAC and DDPG is the inclusion of entropy maximization in the objective. The entropy appears in both the policy and value function. In the policy, it prevents premature convergence of the policy variance (Equation 9). In the value function, it encourages exploration by increase the value of regions of state space that lead to high-entropy behavior (Equation 5). Figure 3a compares how inclusion of this term in the policy and value updates affects the performance, by removing the entropy from each one in turn. As evident from the figure, including the entropy in the policy objective is crucial, while maximizing the entropy as part of the value function is less important, and can even hinder learning in the beginning of training.

**Exploration noise.** Another difference is the exploration noise: the maximum entropy framework naturally gives rise to Boltzmann exploration for SAC, whereas DDPG uses external OU noise (Uhlenbeck & Ornstein, 1930; Lillicrap et al., 2015). For the next experiment, we used a policy with single mixture component and compared Boltzmann exploration to exploration via external OU noise (Figure 3b). We repeated the experiment with the OU noise parameters that we used with

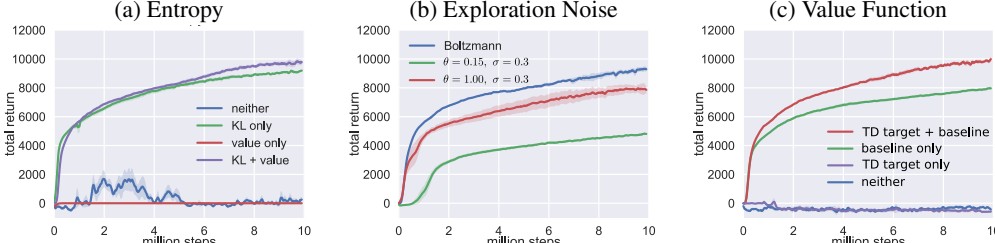

Figure 3: We tested the importance of entropy maximization, Boltzmann exploration, and the separate value network. (a) The label of each curve denotes which entropy terms (KL divergence and/or value function update) were *included* in the policy and value objectives. (b) The blue curve corresponds to the unaltered SAC with Boltzmann exploration, the green and red curves were obtained by using OU exploration noise with the given parameters. (c) Contribution of a separate value function target network in the TD updates and as a policy gradient baseline.

DDPG ($\theta = 0.15$, $\sigma = 0.3$), and parameters that we found worked best for SAC without Boltzmann exploration ($\theta = 1$, $\sigma = 0.3$). The results indicate that Boltzmann exploration outperforms external OU noise.

**Separate value network.** Our method also differs from DDPG in that it uses a separate network to predict the state values, in addition to the Q-values. The value network serves two purposes: it is used to bootstrap the TD updates for learning the Q-function (Equation 7), and it serves as a baseline to reduce variance of the policy gradients (Equation 11). The results in Figure 3c indicates that the value network has an important role in the policy gradient, but only has a minor effect on the TD updates. Nevertheless, the best performance is achieved when the value network is employed for both purposes.

**From SAC to DDPG.** Soft actor-critic closely resembles DDPG with a stochastic actor. To study which of the components that differs between the two algorithms are most important, we evaluated a range of SAC variants by swapping components of SAC with their DDPG counterparts. In Figure 4 we evaluate four versions obtained through incremental modifications: 1) original SAC, 2) replaced likelihood ratio policy gradient estimator with an estimator obtained through the reparametrization trick, 3) replaced the stochastic policy with a deterministic policy and Boltzmann exploration with OU noise, and 4) eliminated the separate value network by using directly the Q-function evaluated at the mean action. The figure suggests two interesting points: First, the reparametrization trick yields somewhat faster and more stable learning compared to the use of likelihood policy gradient estimator, though this

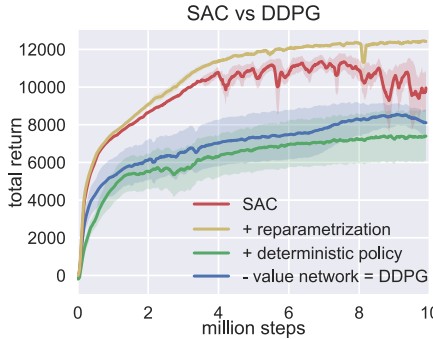

Figure 4: We modified the SAC implementation to make it identical to DDPG by applying a sequence of incremental modifications.

precludes the use of multimodal mixtures of Gaussians. Second, use of a deterministic policy degrades performance substantially in terms of both average return (thick line) and variance across random seeds (shaded region). This observation indicates that entropy maximization, and particularly the use of a stochastic policy, can make off-policy deep RL significantly more robust and efficient compared to algorithms that maximizes the expected return, especially with deterministic policies.

## 6 CONCLUSION

We presented soft actor-critic (SAC), an off-policy maximum entropy deep reinforcement learning algorithm that provides sample-efficient learning while retaining the benefits of entropy maximization and stability. Our theoretical results derive soft policy iteration, which we show to converge to the optimal policy. From this result, we can formulate a soft actor-critic algorithm, and we empirically show that it outperforms state-of-the-art model-free deep RL methods, including the off-policy DDPG algorithm and the on-policy TRPO algorithm. In fact, the sample efficiency of this approach actually exceeds that of DDPG by a substantial margin. Our results suggest that stochastic, entropy maximizing reinforcement learning algorithms can provide a promising avenue for improved robustness and stability, and further exploration of maximum entropy methods, including methods that incorporate second order information (e.g., trust regions (Schulman et al., 2015)) or more expressive policy classes is an exciting avenue for future work.

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

## A  MAXIMUM ENTROPY OBJECTIVE

The exact definition of the discounted maximum entropy objective is complicated by the fact that, when using a discount factor for policy gradient methods, we typically do not discount the state distribution, only the rewards. In that sense, discounted policy gradients typically do not optimize the true discounted objective. Instead, they optimize average reward, with the discount serving to reduce variance, as discussed by Thomas (2014). However, we can define the objective that *is* optimized under a discount factor as

$$J(\pi) = \sum_{t=0}^{\infty} \mathbb{E}_{(\mathbf{s}_s, \mathbf{a}_s) \sim \rho_\pi} \left[ \sum_{l=t}^{\infty} \gamma^{l-t} \mathbb{E}_{\mathbf{s}_l \sim p_\mathbf{s}, \mathbf{a}_l \sim \pi} \left[ r(\mathbf{s}_t, \mathbf{a}_t) + \alpha \mathcal{H}(\pi(\cdot | \mathbf{s}_t)) | \mathbf{s}_t, \mathbf{a}_t \right] \right]. \tag{13}$$

This objective corresponds to maximizing the discounted expected reward and entropy for future states originating from every state-action tuple $(\mathbf{s}_t, \mathbf{a}_t)$ weighted by its probability $\rho_\pi$ under the current policy.

## B  PROOFS

### B.1  LEMMA 1

**Lemma 1** (Soft Policy Evaluation). *Consider the soft Bellman backup operator $\mathcal{T}^\pi$ in Equation 2 and a mapping $Q^0 : \mathcal{S} \times \mathcal{A} \to \mathbb{R}$, and define $Q^{k+1} = \mathcal{T}^\pi Q^k$. Then the sequence $Q^k$ will converge to the soft Q-value of $\pi$ as $k \to \infty$.*

*Proof.* Define the entropy augmented reward as $r_\pi(\mathbf{s}_t, \mathbf{a}_t) \triangleq r(\mathbf{s}_t, \mathbf{a}_t) + \mathbb{E}_{\mathbf{s}_{t+1} \sim p_\mathbf{s}} [\mathcal{H}(\pi(\cdot | \mathbf{s}_{t+1}))]$ and rewrite the update rule as

$$Q(\mathbf{s}_t, \mathbf{a}_t) \leftarrow r_\pi(\mathbf{s}_t, \mathbf{a}_t) + \gamma \mathbb{E}_{\mathbf{s}_{t+1} \sim p_\mathbf{s}, \mathbf{a}_{t+1} \sim \pi} [Q(\mathbf{s}_{t+1}, \mathbf{a}_{t+1})] \tag{14}$$

and apply the standard convergence results for policy evaluation (Sutton & Barto, 1998). $\square$

### B.2  LEMMA 2

**Lemma 2** (Soft Policy Improvement). *Let $\pi_{\mathrm{old}} \in \Pi$ and let $\pi_{\mathrm{new}}$ be the optimizer of the minimization problem defined in Equation 4. Then $Q^{\pi_{\mathrm{new}}}(\mathbf{s}_t, \mathbf{a}_t) \geq Q^{\pi_{\mathrm{old}}}(\mathbf{s}_t, \mathbf{a}_t)$ for all $(\mathbf{s}_t, \mathbf{a}_t) \in \mathcal{S} \times \mathcal{A}$.*

*Proof.* Let $\pi_{\mathrm{old}} \in \Pi$ and let $Q^{\pi_{\mathrm{old}}}$ and $V^{\pi_{\mathrm{old}}}$ be the corresponding soft state-action value and soft state value, and let $\pi_{\mathrm{new}}$ be defined as

$$\pi_{\mathrm{new}}(\cdot | \mathbf{s}_t) = \arg \min_{\pi' \in \Pi} D_{\mathrm{KL}} \left( \pi'(\cdot | \mathbf{s}_t) \, \| \, \exp \left( Q^{\pi_{\mathrm{old}}}(\mathbf{s}_t, \cdot) - \log Z^{\pi_{\mathrm{old}}}(\mathbf{s}_t) \right) \right)$$
$$= \arg \min_{\pi' \in \Pi} J_{\pi_{\mathrm{old}}}(\pi'(\cdot | \mathbf{s}_t)). \tag{15}$$

It must be the case that $J_{\pi_{\mathrm{old}}}(\pi_{\mathrm{new}}(\cdot | \mathbf{s}_t)) \leq J_{\pi_{\mathrm{old}}}(\pi_{\mathrm{old}}(\cdot | \mathbf{s}_t))$, since we can always choose $\pi_{\mathrm{new}} = \pi_{\mathrm{old}} \in \Pi$. Hence

$$\mathbb{E}_{\mathbf{a}_t \sim \pi_{\mathrm{new}}} [\log \pi_{\mathrm{new}}(\mathbf{a}_t | \mathbf{s}_t) - Q^{\pi_{\mathrm{old}}}(\mathbf{s}_t, \mathbf{a}_t) + \log Z^{\pi_{\mathrm{old}}}(\mathbf{s}_t)] \leq \mathbb{E}_{\mathbf{a}_t \sim \pi_{\mathrm{old}}} [\log \pi_{\mathrm{old}}(\mathbf{a}_t | \mathbf{s}_t) - Q^{\pi_{\mathrm{old}}}(\mathbf{s}_t, \mathbf{a}_t) + \log Z^{\pi_{\mathrm{old}}}(\mathbf{s}_t)], \tag{16}$$

and since partition function $Z^{\pi_{\mathrm{old}}}$ depends only on the state, the inequality reduces to

$$\mathbb{E}_{\mathbf{a}_t \sim \pi_{\mathrm{new}}} [Q^{\pi_{\mathrm{old}}}(\mathbf{s}_t, \mathbf{a}_t) - \log \pi_{\mathrm{new}}(\mathbf{a}_t | \mathbf{s}_t)] \geq V^{\pi_{\mathrm{old}}}(\mathbf{s}_t). \tag{17}$$

Next, consider the soft Bellman equation:

$$Q^{\pi_{\mathrm{old}}}(\mathbf{s}_t, \mathbf{a}_t) = r(\mathbf{s}_t, \mathbf{a}_t) + \gamma \mathbb{E}_{\mathbf{s}_{t+1} \sim p_\mathbf{s}} [V^{\pi_{\mathrm{old}}}(\mathbf{s}_{t+1})]$$
$$\leq r(\mathbf{s}_t, \mathbf{a}_t) + \gamma \mathbb{E}_{\mathbf{s}_{t+1} \sim p_\mathbf{s}} \left[ \mathbb{E}_{\mathbf{a}_{t+1} \sim \pi_{\mathrm{new}}} [Q^{\pi_{\mathrm{old}}}(\mathbf{s}_{t+1}, \mathbf{a}_{t+1}) - \log \pi_{\mathrm{new}}(\mathbf{a}_{t+1} | \mathbf{s}_{t+1})] \right]$$
$$\vdots$$
$$\leq Q^{\pi_{\mathrm{new}}}(\mathbf{s}_t, \mathbf{a}_t), \tag{18}$$

where we have repeatedly expanded $Q^{\pi_{\mathrm{old}}}$ on the RHS by applying the soft Bellman equation and the bound in Equation 17. Convergence to $Q^{\pi_{\mathrm{new}}}$ follows from Lemma 1. $\square$

### B.3    THEOREM 1

**Theorem 1** (Soft Policy Iteration).  *Repeated application of soft policy evaluation and soft policy improvement to any $\pi \in \Pi$ converges to a policy $\pi^*(\,\cdot\,|\mathbf{s}_t)$ such that $Q^{\pi^*}(\mathbf{s}_t, \mathbf{a}_t) > Q^{\pi}(\mathbf{s}_t, \mathbf{a}_t)$ for all $\pi \in \Pi$, $\pi \neq \pi^*$ and for all $(\mathbf{s}_t, \mathbf{a}_t) \in \mathcal{S} \times \mathcal{A}$.*

*Proof.*  Let $\pi_i$ be the policy at iteration $i$. By Lemma 2, the sequence $Q^{\pi_i}$ is monotonically increasing. Since $Q^{\pi}$ is bounded above for $\pi \in \Pi$ (both the reward and entropy are bounded), the sequence converges to some $\pi^*$. We will still need to show that $\pi^*$ is indeed optimal. At convergence, it must be case that $J_{\pi^*}(\pi^*(\,\cdot\,|\mathbf{s}_t)) < J_{\pi^*}(\pi(\,\cdot\,|\mathbf{s}_t))$ for all $\pi \in \Pi$, $\pi \neq \pi^*$. Using the same iterative argument as in the proof of Lemma 2, we get $Q^{\pi^*}(\mathbf{s}_t, \mathbf{a}_t) > Q^{\pi}(\mathbf{s}_t, \mathbf{a}_t)$ for all $(\mathbf{s}_t, \mathbf{a}_t) \in \mathcal{S} \times \mathcal{A}$, that is, the soft value of any other policy in $\Pi$ is lower than that of the converged policy. Hence $\pi^*$ is optimal in $\Pi$.     □

## C    ENFORCING ACTION BOUNDS

We use an unbounded GMM as the action distribution. However, in practice, the actions needs to be bounded to a finite interval. To that end, we apply an invertible squashing function ($\tanh$) to the GMM samples, and employ the change of variables formula to compute the likelihoods of the bounded actions. In the other words, let $\mathbf{u} \in \mathbb{R}^D$ be a random variable and $\mu(\mathbf{u}|\mathbf{s})$ the corresponding density with infinite support. Then $\mathbf{a} = \tanh(\mathbf{u})$, where $\tanh$ is applied elementwise, is a random variable with support in $(-1, 1)$ with a density given by

$$\pi(\mathbf{a}|\mathbf{s}) = \mu(\mathbf{u}|\mathbf{s}) \left| \det\left(\frac{d\mathbf{a}}{d\mathbf{u}}\right) \right|^{-1}. \tag{19}$$

Since the Jacobian $d\mathbf{a}/d\mathbf{u} = \mathrm{diag}(1 - \tanh^2(\mathbf{u}))$ is diagonal, the log-likelihood has a simple form

$$\log \pi(\mathbf{a}|\mathbf{s}) = \log \mu(\mathbf{u}|\mathbf{s}) - \sum_{i=1}^{D} \log\left(1 - \tanh^2(u_i)\right), \tag{20}$$

where $u_i$ is the $i^{\mathrm{th}}$ element of $\mathbf{u}$.

## D    SENSITIVITY TO HYPERPARAMETERS

This section discusses the sensitivity to hyperparameters. All of the following experiments are on the HalfCheetah-v1 benchmark, and although we found these results to be representative in most cases, they may differ between the environments. For other environments, we only swept over the reward scale only.

**Reward scale.**    Soft actor-critic is particularly sensitive to the reward magnitude, because it serves the role of the temperature of the energy-based optimal policy and thus controls its stochasticity. Figure 5a shows how learning performance changes when the reward scale is varied: For small reward magnitudes, the policy becomes nearly uniform, and consequently the fails to exploit the reward signal, resulting in substantial degradation of performance. For large reward magnitudes, the model learns quickly at first, but the policy quickly becomes nearly deterministic, leading to poor exploration and subsequent failure due to lack of adequate exploration. With the right reward magnitude, the model balances exploration and exploitation, leading to fast and stable learning. In practice, we found reward scale to be the only hyperparameter that requires substantial tuning, and since this parameter has a natural interpretation in the maximum entropy framework, this provides good intuition for how to adjust this parameter.

**Mixture components.**    We experimented with different numbers of mixture components in the Gaussian mixture policy (Figure 5b). We found that the number of components did not affect the learning performance very much, though larger numbers of components consistently attained somewhat better results.

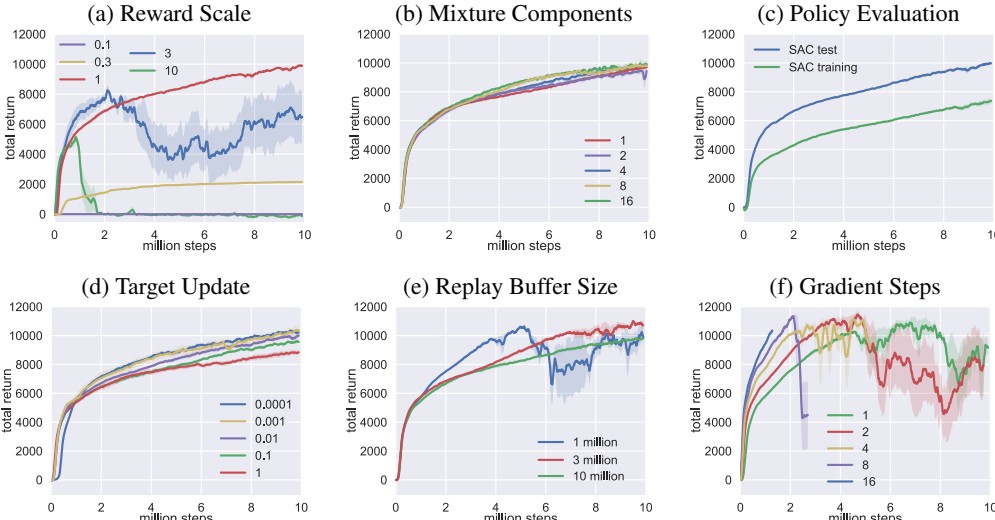

Figure 5: Sensitivity of soft actor-critic to selected hyperparameters in the HalfCheetah-v1 task. See Appendix D for more detailed analysis.

**Policy evaluation.**    Since SAC converges to stochastic policies, it is generally beneficial to make the final policy deterministic at the end for best performance. For evaluation, we approximate the maximum a posteriori action by choosing the action that maximizes the Q-function among the mixture component means. Figure 5c compares training returns to test returns obtained with this strategy. The test returns are substantially better. It should be noted that all of the training curves depict the sum of rewards, which is different from the objective optimized by SAC and SQL which both maximize also the entropy.

**Target network update.**    In Figure 5d we varied the smoothing constant, $\tau$, for the target value network updates. A value of one corresponds to a hard update where the weights are copied directly at every iteration, and values close to zero correspond to exponentially moving average of the network weights. Interestingly, SAC is relatively insensitive to the smoothing constant: smaller $\tau$ tends to yield higher performance at the end with the cost of marginally slower progress in the beginning, but the difference is small. In prior work, the main use of the target network was to stabilize training (Mnih et al., 2015; Lillicrap et al., 2015), but interestingly, SAC is able to learn stably also when effectively no separate target network is used, but instead the weights are copied directly ($\tau = 1$), although at the cost of minor degradation of performance.

**Replay buffer size.**    Next, we experimented with the size of the experience replay buffer (Figure 5e), which we found to be important for environments where the optimal policy becomes nearly deterministic at convergence, such as HalfCheetah-v1. For such environments, the exploration noise becomes negligible at the end, making the content of the replay buffer less diverse and resulting in overfitting and instability. Use of a higher capacity buffer solves this problem by allowing the method to remember the failures from the beginning of learning, but slows down learning by allocating some of the network capacity to modeling the suboptimal initial experience. Note that a return of just 4800 is considered as the threshold level for solving this benchmark task and that the performance in all cases is well beyond the threshold. We used a buffer size of 10 million samples for HalfCheetah-v1, and 1 million samples for the other environments.

**Gradient steps per time step.**    Finally, we experimented with the number of actor and critic gradient steps per time step of the algorithm (Figure 5f). Prior work has observed that increasing the number of gradient steps for DDPG tends to improve sample efficiency (Gu et al., 2017; Popov et al., 2017), but only up to a point. We found 4 gradient updates per time step to be optimal for DDPG, before we saw degradation of performance. SAC is able to effectively benefit from substantially larger numbers of gradient updates in most environments, with performance increasing steadily until 16 gradient updates per step. In some environments, such as Humanoid, we did not observe as much benefit. The ability to take multiple gradient steps without negatively affecting the algorithm is important especially for learning in the real world, where experience collection is the bottleneck,

and it is typical to run the learning script asynchronously in a separate thread. In this experiment, we used a replay buffer size of 1 million samples, and therefore the performance suddenly drops after reaching a return of approximately 10 thousand due to the reason discussed in the previous section.

## E   HYPERPARAMETERS

Table 1 lists the common SAC parameters used in the comparative evaluation in Figure 1, and Table 2 lists the parameters that varied across the environments.

Table 1: Shared Parameters

| Parameter | Symbol | Value |
|---|---|---|
| learning rate | | $3 \cdot 10^{-4}$ |
| discount | $\gamma$ | 0.99 |
| target smoothing coefficient | $\tau$ | 0.01 |
| number of mixture components | K | 4 |
| number of layers (3 networks) | | 2 |
| number of hidden units per layer | | 128 |

Table 2: Environment Specific Parameters

| Environment | DoFs | Reward Scale | Gradient Steps | Replay Pool Size |
|---|---|---|---|---|
| Swimmer (rllab) | 2 | 100 | 4 | $10^6$ |
| Hopper-v1 | 3 | 1 | 4 | $10^6$ |
| HalfCheetah-v1 | 6 | 1 | 4 | $10^7$ |
| Walker2d-v1 | 6 | 3 | 4 | $10^6$ |
| Ant-v1 | 8 | 3 | 1 | $10^6$ |
| Humanoid (rllab) | 21 | 3 | 1 | $10^6$ |

