# OpenReview forum: "Soft Actor-Critic: Off-Policy Maximum Entropy Deep Reinforcement Learning with a Stochastic Actor"
_ICLR.cc/2018/Conference — Invite to Workshop Track_

### Official Review · AnonReviewer1 · 2017-11-27
**Issues with novelty and some correctness issues**

**Rating:** 3
**Confidence:** 4

**Review:**

This paper proposes a soft actor-critic method aiming at lowering sample complexity and achieving a new convergence guarantee. However, the current paper has some correctness issues, is missing some related work and lacks a clear statement of innovation.

The first issue is that augmenting reward by adding an entropy term to the original RL objective is not clearly innovative. The connections, and improvements upon, other approaches need to be made more clear. In particular, the connection to the work by Haarnoja is unclear. There is this statement: “Although the soft Q-learning algorithm proposed by Haarnoja et al. (2017) has a value function and actor network, it is not a true actor-critic algorithm: the Q- function is estimating the optimal Q-function, and the actor does not directly affect the Q-function except through the data distribution. Hence, Haarnoja et al. (2017) motivates the actor network as an approximate sampler, rather than the actor in an actor-critic algorithm. Crucially, the convergence of this method hinges on how well this sampler approximates the true posterior. In contrast, we prove that our method converges to the optimal policy from a given policy class, regardless of the policy parameterization.” The last sentence suggests that the key difference is that any policy parameterization can be used, making the previous sentences less clear. Is the key extension on the proof, and so on the use of the projection with the KL-divergence?

Further, there is a missing connection to the paper “Guided policy search”, Levine and Koltun. Though it is a different framework, it clearly mentioned it uses the augmented reward to learn the sub-optimal policies (for differential dynamic program). The DDPG paper mentioned that DDPG can be also used within the GPS framework. That work is different, but a discussion should nonetheless be included about connections.

If the key novelty in this work is an extension on the theory, to allow any policy parameterization, and empirical results demonstrating improved performance over Haarnoja et al., there appear to be correctness issues in both, as laid out below.

The key novelty in the theory seems to be to use a projection onto the space of policies, using a KL divergence. There are, however, currently too many unclear or misspecified steps to verify correctness.
1. The definition of pinew in Equation (6) is for one specific state s_t; shouldn’t this be across all states? If it is for one state, then E_{pinew} makes sense, since pi is only specified as a conditional distribution (not a joint distribution); if it is supposed to be expected value across all states, then what is E_{pinew}? Is it with the stationary distribution of pinew?

2. The proof for Lemma 1 is hard to follow, because Z is not defined. I mostly was able to guess, based on Haarnoja et al., but the step between (18) and (19) where E_{pinew} [Zpiold] -  E_{piold} [Zpiold] is dropped is unclear to me. Zpiold does not depend on actions, so if the expectation is only w.r.t. to the action, then it cancels. This goes back to point 1, where it wouldn’t make much sense for the KL to only depend on actions. In fact, if pinew has to be computed separately for each state, then we are really back to tabular policies.

3. “There is no need in principle to include a separate function approximator for the state value, since it is related to the Q-function and policy according to Qθ (st , at ) − log πφ (at |st )” This is not true, since you rely on the fact that you have separate network parameters to get an unbiased gradient estimate in (10).

The experimental results also appear to have some correctness issues.
1. For the claim that the algorithm does better, this is also difficult to gauge because the graphs are unclear. In particular, it is not explained why the lines end early. How were multiple gradients incorporated into the update? Did you wait 4 (or 16) steps until computing a gradient update? This might explain why the lines end early, but then invalidates how the lines are drawn. Rather, the lines should be extended, where each point is plotted each 4 (or 16) steps. Doing this would remove the effect that the lines with 4 or 16 seem to learn faster (but really are just plotted on a different x-axis).

2. There is a lack of experimental details. This includes missing details about neural network architectures used by each algorithm, parameter tuning details, how multiple gradients are used, etc. This omission makes the experiments not reproducible.

3. Although DDPG is claimed to be very sensitive to parameter changes, and the proposed algorithm is more stable, there is no parameter sensitivity results showed.

Minor comments:
1. Graph font is much too small.
2. Typo in (10), should be V(s_{t+1})
3. Because the proof of Lemma 1 is so straightforward (just redefining reward), it would be better to actually spell it out, give a definition of entropy, etc.

---

> ### Author Response · Authors · 2018-01-04
> **Response**
>
> Thank you for your comments and feedback. We have added a number of additional experiments to address each of your concerns about the empirical results, and revised the paper to address all of your concerns regarding the theoretical results. We summarize these below.
>
> As R3 notes, our method is novel when considered in the context of recent work in deep reinforcement learning. While the notion of entropy regularization is certainly not new (nor do we claim this), the particular method we propose is novel, and to our knowledge no prior work has proposed an off-policy actor-critic algorithm for optimizing the maximum entropy RL objective for continuous control. The empirical results show that this method substantially outperforms the previous state of the art in terms of sample efficiency on a range of very challenging continuous control tasks. We believe that state-of-the-art results on widely accepted benchmark tasks are of significant interest to the community, and merit publication in ICLR.
>
> To address your concerns, we have revised the introduction to better communicate the contribution of our paper. Prior methods for maximum entropy policies have been formulated as Q-learning methods that learn the Q-function of the optimal policy directly, even though the optimal policy in continuous domains is intractable and the optimal behavior might not be reproducible. Another benefit of our formulation is that the practical approximation, the soft actor-critic algorithm, is simple to implement and does not rely on any biased approximations (such as estimation of the optimal value function in soft Q-learning) or approximate inference of the optimal policy (such as Stein variational gradient descent in soft Q-learning) which increases the time complexity. We have also cited the guided policy search work in the related work section.
>
> Responses to the comments on the theory:
> 1. The minimization is indeed performed for each state independently. We have added clarification before Equation 4 (old Equation 6) to make it explicit.
>
> 2. Thank you for pointing out the readability of the proof was insufficient. We have now revised the proofs to improve their clarity. To answer your specific question, Z is the partition function that, as you guessed, does not depend on actions, and therefore cancels out, which we now state explicitly in the proof.
>
> 3. In fact, it is possible to estimate the value function in Equation 7 (was Equation 10) with Q - log \pi evaluated at an action sampled from the current policy without introducing a bias. We have revised the second paragraph in Section 4.2 to better explain this point. We have also included an ablation in the experiment section where we estimate the value using the Q-function and policy directly, and we did not observe any significant difference in performance, but we found that the value function has an important role as a baseline for the off-policy policy gradient.
>
> Responses to the comment on the experimental results:
> 1. We have addressed your concerns about the results. Indeed each of the experiments should have been run to convergence, we were unfortunately unable to do this for this submission due to time constraints. We have now updated the paper to include the full results as you requested. The current version of the paper has updated results, and now includes a soft Q-learning baseline (see Figure 1 on page 7), and we will also include NAF comparison in the final version of our paper. We took multiple gradient steps between sampling new evidence from the environment. Since the gradients are computed using samples from the replay buffer, they are not considered as additional steps in the learning curves, where the x-axes correspond to the number of environment steps. However, taking multiple gradient steps makes each step slower in terms of the wall-clock time--hence the experiment with a large number of gradient steps end earlier.
>
> 2. We have added experimental details to Appendix C (how to apply our method to bounded action domains) and Appendix E (list of all hyperparameters we used in the experiments). We will also release a link to our code with the final version of our paper for reproducibility.
>
> 3. We have added sensitivity study over the most important hyperparameters in Appendix D.

---

### Official Review · AnonReviewer3 · 2017-11-28
**Impressive empirical results**

**Rating:** 7
**Confidence:** 4

**Review:**

The paper presents an off-policy actor-critic method for learning a stochastic policy with entropy regularization. It is a direct extension of maximum entropy reinforcement learning for Q-learning (recently called soft-Q learning), and named soft actor-critic (SAC). Empirically SAC is shown to outperform DDPG significantly in terms of stability and sample efficiency, and can solve relatively difficult tasks that previously only on-policy (or hybrid on-policy/off-policy) method such as TRPO/PPO can solve stably. Besides entropy regularization, it also introduces multi-modal policy parameterization through mixture of Gaussians that enables diverse, on-policy exploration.

The main appeal of the paper is the strong empirical performance of this new off-policy method in continuous action benchmarks. Several design choices could be the key, so it is encouraged to provide more ablation studies on these, which would be highly valuable for the community. In particular,

- Amortization of Q and \pi through fitting state value function

- On-policy exploration vs OU process based off-policy exploration

- Mixture vs non-mixture-based stochastic policy

- SAC vs soft Q-learning

Another valuable discussion to be had is the stability of off-policy algorithm comparing Q-learning versus actor-critic method.

Pros:

- Simple off-policy algorithm that achieves significantly better performance than existing off-policy baseline algorithms

- It allows on-policy exploration in off-policy learning, partially thanks to entropy regularization that prevents variance from shrinking to 0. It could be considered a major success of off-policy algorithm that removes heuristic exploration noise.

Cons:

- Method is relatively simple extension from existing work in maximum entropy reinforcement learning. It is unclear what aspects lead to significant improvements in performance due to insufficient ablation studies.


Other question:

- Above Eq. 7 it discusses that fitting a state value function wrt Q and \pi is shown to improve the stability significantly. Is this comparison with directly estimating state value using finite samples? If so, is the primary instability due to variance of the estimate, which can be avoided by drawing a lot of samples or do full integration (still reasonably tractable for finite mixture model)? Or, is the instability from elsewhere? By having SGD-based fitting of state value function, it appears to simulate slowly changing target values (similar role as target networks). If so, could a similar technique be used with DDPG and get more stable performance?

---

> ### Author Response · Authors · 2018-01-04
> **Response**
>
> Thank you for you constructive and useful suggestions. We have extended the experiment section to include all the suggested ablations and also added discussion and experiments regarding the sensitivity to hyperparameters in the appendix. In short, use of a separate value network has only a minor contribution to the stability of learning the Q-network, but it is crucial to be used as a baseline for the policy gradients and therefore cannot be excluded from the algorithm (see Figure 3 (c) in Section 5.2). One of the benefits of our formulation is that we can estimate the state value with a single action sample. This is possible since we estimate the value of the current policy instead of the value of the optimal policy as in soft Q-learning, where the estimation requires computing “log-sum-exp.” In our version, we only need to evaluate the expectation (“sum”), which can be estimated with a single sample without introducing a bias. In DDPG, the expectation is replaced with the evaluation of the Q-function at the current policy mean, which we found to be less robust and potentially the most important factor that makes our proposed algorithm work well.

---

> > ### Comment · AnonReviewer3 · 2018-01-08
> > **I've read the authors' rebuttal, and I stick with my original assessment score.**
> >
> > They have added requested ablation studies in the main text (5.2) and some hyper-parameter sensitivity experiments in the appendix, with sufficient discussions on the results. While the entropy regularized actor critic is a direct extension on MaxEnt RL/soft Q-learning etc., the algorithm has shown impressive results and the follow-up studies are done sufficiently. I recommend for acceptance.

---

### Official Review · AnonReviewer2 · 2017-12-01
**The paper considers the problem of Deep RL in continuous-action domains. It implements the well-studied idea of RL with Entropy bonus. The results on the control suit looks very promising, though the paper does not compare with the state-of-the-art variants of baseline. Also the implementation details of the algorithm is not completely provided. So it is difficult to fully assess  the empirical results.**

**Rating:** 5
**Confidence:** 4

**Review:**

Quality and clarity:

It seems that the authors can do a better job to improve the readability of the paper and its conciseness. The current structure of paper seems a bit suboptimal to me. The first 6.5 page of the paper is used to explain the idea of RL with entropy  reward and how it can be extended to the case of parametrized value function and policy and then the whole experimental results  is packed  in only 1 page.  I think the paper could be organized in a more balanced way  by providing a more detailed description and analysis of the numerical results, especially given the fact that in my opinion this is the main strength of the paper.  Finally some of the claims made in this paper is not really justified. For instance "Prior deep RL methods based on this framework have been formulated as either off-policy Q-learning, or on-policy policy gradient methods" not true, e.g., look at  Deepmind AI recent work:  https://arxiv.org/abs/1704.04651.

Originality and novelty:

I think much of the ideas considered in this paper is already explored in previous work as it is acknowledged  in the paper.  However some of the techniques such as the way the policy is  represented  and the way the policy gradient formulation is approximated  seems to be novel in the context of Deep RL though again these ideas have been explored in the literature of control and RL extensively.

Significance:

I think the improvement on baseline in control suites is very impressive the problem is that the details of the implementation of algorithm e.g. architecture of neural network size of memory replay, the schedule the target network  is implemented is not sufficiently explained in the paper so it is  difficult to evaluate these results. Also the paper only compares with the original version of the baseline algorithms. These algorithms are improved since then and the new more efficient algorithms such as distributional policy gradients and NAF have been developed. So it would help to have a better understanding of this result if the paper compares with the state-of-the-art baselines.

Minor:
for some reason different algorithms have ran for different number of steps which is a bit confusing. would be great if this is fixed in the future version.

---

> ### Author Response · Authors · 2018-01-04
> **Response**
>
> Thank you for your constructive comments and suggestions. We believe that our work does indeed compare to the state-of-the-art methods in terms of sample efficiency, and we have added a comparison to soft Q-learning (see the revised Figure 1 on page 7) and will add also NAF in the final to attempt to address your concerns. If there are specific other prior methods that the reviewer would like to see a comparison to, we would be happy to add them in the final. We address specific points raised by the reviewer below.
>
> To address your concerns, we have extended the experiment section to include soft Q-learning as a baseline and will include NAF in the final. Distributional policy gradients is a concurrent submission to ICLR, and it therefore does not seem appropriate to require a comparison to this method, though we would be happy to discuss it in the final.
>
> Our paper  has two major contributions: first, a novel theoretical framework, soft policy iteration, that is generally applicable for optimizing maximum entropy objectives, and second, a practical soft actor-critic algorithm that makes use of the theory and achieves state-of-the-art results in continuous benchmark tests. We think both of the contributions are of comparable importance, and therefore we have allocated a large part of the paper to the derivation and discussion of the theoretical framework.
>
> Regarding novelty: we are not aware of prior works that propose off-policy actor-critic algorithms within the maximum entropy framework for continuous control tasks, though if the reviewer has any suggestions for works of this kind, we would be happy to cite and discuss them. To our knowledge, the results reported in our experiments substantially improve on the state of the art (DDPG) in terms of sample efficiency, often by a very large margin. We believe that substantial improvements over the state of the art on sample efficiency, which is a crucial problem in deep RL, are of sufficient interest to the community to merit publication in ICLR.
>
> To address your comment regarding the statement “Prior deep RL methods based on this framework have been formulated as either off-policy Q-learning, or on-policy policy gradient methods.” Note that this refers specifically to maximum entropy algorithms. To our knowledge, the only deep RL methods that optimize the entropy augmented objective are based on off-policy soft Q-learning (and similar prior methods that work in discrete domains), or on-policy policy gradients. Methods such as Gruslys et al. (referenced in your review) do not optimize the same objective, but rather use an entropy regularizer. We have revised the paper to address this and cited Gruslys et al. in the related work section, though we cannot compare to that method directly since it addresses discrete-action problems.

---

### Decision · Program_Chairs · 2018-01-29
**ICLR 2018 Conference Acceptance Decision**

**Decision:**

Invite to Workshop Track

**Comment:**

The reviewers agree that the results are promising and there are some interesting and novel aspect to the formulation. However, two of the reviews have raised concerns regarding the exposition and the discussion of previous work. The paper benefits from a detailed description of soft Q-learning, PCL, and off-policy actor-critic algorithms, and how SAC is different from those. Instead of differentiating against previous work by saying soft Q-learning and PCL are not actor-critic algorithms, discuss the similarities and differences and present empirical evaluation.